# Accumulated Vitrified Embryos Could Be a Method for Increasing Pregnancy Rates in Patients with Poor Ovarian Response

**DOI:** 10.3390/jcm11174940

**Published:** 2022-08-23

**Authors:** Jieun Shin, Hwang Kwon, Dong Hee Choi, Chan Park, Ji Hyang Kim, Jeehyun Kim, Youn-Jung Kang, Hwa Seon Koo

**Affiliations:** 1Department of Obstetrics and Gynecology, CHA Fertility Center Bundang, 59, Yatap-ro, Bundang-gu, Seongnam-si 13496, Gyeonggi-do, Korea; 2Department of Biochemistry, Research Institute for Basic Medical Science, School of Medicine, CHA University, 335 Pangyo-ro, Bundang-gu, Seongnam-si 13488, Gyeonggi-do, Korea

**Keywords:** poor ovarian response, embryo accumulation, assisted reproduction technique, retrospective study, embryo transfer

## Abstract

We aimed to assess the efficacy of accumulated embryo transfer (ACC-ET) through several controlled ovarian hyperstimulation (COS) cycles to increase the rates of pregnancy in patients with poor ovarian response (POR). We retrospectively reviewed the medical records of 588 patients with POR under 43-years old who underwent embryo transfer from January 2010 to December 2015. We compared the pregnancy rate (PR), clinical pregnancy rate (CPR), and live birth rate (LBR) between ACC-ET (frozen-thawed: 47; fresh + frozen-thawed: 24) group (n = 71) and fresh ET groups (n = 517). Characteristics of ACC-ET patients were similar to those of fresh ET groups (Age: 38.1 ± 3.5 vs. 38.2 ± 3.7, *p* = 0.88; Anti Müllerian Hormone (AMH; ng/mL): 0.5 ± 0.4 vs. 0.6 ± 0.6, *p* = 0.38; follicle stimulating hormone (FSH: mIU/mL): 11.9 ± 8.0 vs. 10.8 ± 9.0, *p* = 0.35). The total number of transferred embryos (3.1 ± 0.9 vs. 1.5 ± 0.7, *p* = 0.00), PR (29.6% (21/71) vs. 18.8% (97/517), *p* = 0.040), and CPR (23.5% (16/68) vs. 14.0% (71/508) *p* = 0.047) were significantly higher in the ACC-ET group than in the fresh ET group. In addition, PR, CPR, and LBR increased with the number of ET in the fresh ET group. However, there were no significant differences observed in LBR between ACC-ET and fresh ET groups (14.9% (10/67) vs. 9.8% (50/508), *p* = 0.203). From our knowledge, there is no clinical evidence reported to prove that transfer of multiple embryos of adequate quality obtained through multiple cycles can compensate for the limited number of retrieved oocytes from POR patients. We concluded that ACC-ET from several COS cycles could be an alternative method to increase PR and CPR in <43-year-old patients with POR.

## 1. Introduction

Assisted reproduction technique (ART) is a common therapeutic method for overcoming infertility. With technological advances, the pregnancy rate following ART has significantly improved. However, poor ovarian response (POR), despite an adequate use of gonadotropin, is still a difficult problem to solve in patients receiving in-vitro fertilization (IVF). Although POR has been defined by many researchers [1,2], the Bologna criteria as described by Ferraretti et al. [3] is the most widely used. However, many researchers [4] have proposed the need for a more detailed definition of POR. The incidence of POR varies between 9 and 24% [5]. In most POR patients, it is known that two or more oocytes, though suboptimal, can be harvested through controlled ovarian hyperstimulation (COS). However, patients who have failed hyperstimulation with a high dose of hormonal treatments might obtain one or more oocytes through natural or modified natural protocol. Kedem et al. [6] showed that according to the Bologna criteria, the pregnancy rate of POR patients who went through the modified natural cycle was very low (0.9% per cycle). Moreover, Polyzos et al. [7] reported a live birth rate of 2.6% per cycle in the natural cycle IVF. The application of natural or modified natural cycle IVF in poor responders does not provide substantial benefits. In poor responders, oocyte donation or adoption may be the only alternative methods after failure of natural or modified natural cycle IVF [6,7]. However, most patients want to get pregnant with their own gamete in some nations; it is difficult for POR patients to receive donated oocytes due to strict bioethics legislation. Various protocols, including an increase in the starting dose of the gonadotropin [4,8,9], addition of recombinant luteinizing hormone (LH) [10,11], combination clomiphene citrate (CC) with gonadotropin [12,13], addition of growth hormone (GH) to a controlled ovarian stimulation protocol for patients with POR [14], and DHEA adjuvant therapy [15], have been tested in attempts to increase the pregnancy rates in patients with POR; however, none of the strategies has proved effective among the poor responders.

Infertility physicians accumulate vitrified embryos produced by several rounds of natural or modified natural protocol and empirically transfer the accumulated thawed embryos to overcome the low pregnancy rate, expecting that an increased number of transferred embryos would improve the pregnancy rate [16]. To the best of our knowledge, there is a lack of evidence proving that the transfer of one or more embryos increases the pregnancy rates in poor responders. If this is clinically evidenced despite the high costs for the multiple oocyte retrievals, it could provide a therapeutic strategy to improve the pregnancy rates of poor responders. Therefore, the aim of current study is to evaluate the efficacy of accumulation of embryo transfer (ACC-ET) from several ovarian stimulations (or IVF cycles) compared to fresh ET in patients with POR under 43 years focusing on the number of transferred embryos.

## 2. Materials and Methods

### 2.1. Study Design and Patients

This retrospective study was approved by the Institutional Review Board of our hospital (IRB number: 2018-11-015-001). We reviewed the medical records of all women with POR who underwent ET between January 2010 and November 2015 at our fertility center and divided them into accumulated group (ACC-ET) (n = 71) and fresh ET group (n = 517). We included only POR patients less than 43 years who fulfilled two out of the three Bologna criteria ((1) advanced maternal age or any other risk for POR (retrieval of four of less oocytes was accepted as a cut off point for POR), (2) a previous poor response to stimulation, and (3) abnormal test of ovarian reserve). The ACC-ET group included frozen-thawed ET (n = 47) and combined fresh and frozen-thawed ET (n = 24) cycles.

### 2.2. COS Protocol and Fresh ET

We used the natural or modified natural cycle for follicular growth. An antagonist (Cetrotide^®^ 0.25 mg per day; Merck Serono, Feltham, UK) was injected subcutaneously when a leading follicle reached 14 mm diameter in the modified natural cycle. Human chorionic gonadotropin (hCG) (Ovidrel^®^; Merck Serono, Roma, Italy) was administered when a leading follicle with a mean diameter ≥ 18 mm was observed during ultrasonography in natural or modified natural cycle. Oocytes, retrieved 35–36 h after hCG injection, were subsequently fertilized by IVF or intra-cytoplasmic sperm injection (ICSI). After 3 days, embryo transfer was done. The same protocol was used in the ACC-ET group.

### 2.3. Accumulation of Vitrified Embryos and Thawing Procedure

The vitrification method was used for freezing the embryos as described in previous studies [17,18]. This procedure of oocyte retrieval, fertilization, and verification of embryos was repeated until more than three vitrified embryos were obtained. For thawing, embryos on EM gold grids were sequentially transferred to thawing solution containing 1.0, 0.5, 0.25, or 0.125 M sucrose in dPBS at 37 °C at intervals of 2.5 min. Following four to six washes with 20% SSS in dPBS, the embryos were placed in the culture medium as previously described [17,18].

### 2.4. ACC-ET (Frozen Thawed ET)

In patients with a regular menstrual cycle, the thawed embryos were transferred on post-ovulation day 3 through follow up of a natural cycle without hormonal addition. In an artificial cycle, estradiol valerate (Progynova^®^; Schering, Seoul, South Korea) was administered from menstruation day 3 and the thawed embryos were transferred on the third day following administration of progesterone (Utrogestan^®^ 200 mg Vaginal Capsule; Capsugel, Illkirch-Graffenstaden, France) when endometrial thickness ≥ 8 mm was observed.

### 2.5. ACC-ET (Combined Fresh and Frozen Thawed ET)

After more than two embryos were vitrified by “accumulation of vitrified embryos” method, the natural or modified natural cycle commenced. Oocyte retrieval and fertilization were performed as described above for the natural or modified natural cycle IVF method. Subsequently, fresh embryos were transferred together with thawed embryos on post-ovulation day 3 in natural and modified natural cycle IVF.

### 2.6. Outcome Measure

For comparison of pregnancy outcome in the two groups, the following factors were recorded: Pregnancy was defined as b-hCG ≥ 10 mIU/mL. Clinical pregnancy was defined as the presence of a gestational sac with fetal heart activity. Live birth was defined as gestational age over 28 weeks. Data were expressed as mean ± SD or percentage, as appropriate. Student’s *t*-test, χ^2^ test, and Fisher’s exact test were used to determine statistical significance. A *p* value < 0.05 was considered statistically significant. SPSS version 22 (IBM Corporation, Armonk, NY, USA) was used for statistical analysis.

## 3. Results

### 3.1. ACC-ET vs. Fresh ET

Patients’ characteristics, including age: 38.2 ± 3.7 vs. 38.1 ± 3.5, *p* = 0.88; AMH: 0.6 ± 0.6vs. 0.5 ± 0.4, *p* = 0.38; and FSH: 10.8 ± 9.0 vs. 11.9 ± 8.0, *p* = 0.35 were similar in both fresh ET and ACC-ET groups (Table 1). In the ACC-ET group, the cancellation rate was 18.8% (45/239), and the average number of oocyte retrieval attempts per patient was 3.3 (Table 2). The number of retrieved oocytes per ET was significantly higher (*p* = 0.00), whereas the number of retrieved oocytes per retrieval was significantly lower in the ACC-ET group (*p* = 0.00). The maturation rates (%) of retrieved oocytes were similar between the two groups (*p* = 0.35), whereas fertilization rates were significantly higher in the fresh ET group (*p* = 0.009). The numbers of transferred embryos (*p* = 0.00), pregnancy rates (PR) (*p* = 0.040), and clinical pregnancy rates (CPR) (*p* = 0.047) were significantly higher in the ACC-ET group. However, the rates of good quality transferred embryos were significantly lower in the ACC-ET group (*p* = 0.00). Moreover, live birth rates (LBR) were higher in the ACC-ET group without statistical significance (*p* = 0.203) (Table 2). A greater number of transferred embryos resulted in higher pregnancy rates even if the quality of embryo transferred was lower.

### 3.2. Subgroup Analysis

Patients’ characteristics between frozen thawed ACC-ET and fresh ET and between frozen thawed + fresh ACC-ET and fresh ET were similar (Table 1).

#### 3.2.1. Frozen Thawed ACC-ET vs. Fresh ET

The average number of oocyte retrieval attempts per patient was 3.4 (160/47). The number of retrieved oocytes per retrieval was significantly lower in the frozen thawed ACC-ET group (*p* = 0.025). The fertilization rates (*p* = 0.00) and good quality embryo rates (*p* = 0.00) were significantly higher in the fresh ET group. The number of transferred embryos was significantly higher in the frozen thawed ACC-ET group (*p* = 0.00). However, there were no significant differences in PR (*p* = 0.119), CPR (*p* = 0.189), and LBR (*p* = 0.277) between the two groups (Table 2).

#### 3.2.2. Frozen Thawed + Fresh ACC-ET and Fresh ET

The number of retrieved oocytes per retrieval was significantly lower in the frozen thawed + fresh ACC-ET group (1.25 ± 0.6 vs. 2.1 ± 0.8, *p* = 0.000). The average number of oocyte retrieval attempts per patient was 3.3 (79/24). Fertilization rates were similar between the two groups (88.2 ± 15.3 vs. 87.6 ± 28.3, *p* = 0.90). The maturation rates in the retrieved oocytes were significantly higher than in the fresh ET group (46.1 ± 27.8 vs. 61.1 ± 36.5, *p* = 0.017; 43.8 ± 30.0 vs. 65.4 ± 41.0, *p* = 0.00). The number of transferred embryos was significantly higher in the frozen thawed ACC-ET group (2.8 ± 0.7 vs. 1.5 ± 0.7, *p* = 0.00). The good quality embryo rates were similar between frozen thawed + fresh ACC-ET and fresh ET groups (56.5 ± 29.9 vs. 65.4 ± 41.0, *p* = 0.17). There were no significant differences in PR, CPR, and LBR between two groups (29.2 (7/24) % vs. 18.8 (97/517), *p* = 0.291; 25.0% (6/24) vs. 14.0% (71/508), *p* = 0.240; 12.5% (3/24) vs. 9.8% (50/508), *p* = 0.708 (Table 2).

#### 3.2.3. Fresh ET

The PR, CPR, and LBR were increased according to the number of transferred embryos (Table 3).

## 4. Discussion

Based on our findings, it can be assumed that increasing the number of embryo transfers through ACC-ET would increase the pregnancy rates in POR patients. POR is a difficult issue to resolve even with advances in technology for infertility treatments. Many therapeutic options have been proposed to overcome POR, but none have proven efficacy [7,9,11]. In POR patients, natural or modified natural protocol are often used because the number of retrieved oocytes is similar to conventional COS, but the cost associated with COS is much higher. With this method, single embryo transfer is opted for IVF. Many studies have reported that single embryo transfer does not lower LBR in patients under 40 or 40–45 years of age with good prognosis [19,20]. However, there is no study yet that compares the results of single versus double embryo transfer in POR patients.

With the stabilization of the thawing technique, it is now possible to transfer embryos collected over multiple cycles. It is thought that selective transfer of good quality embryos would increase the pregnancy rates in POR patients and reduce the discrepancy between the endometrium and the embryo due to hyper-response. Based on this theory and experience, transfer of multiple embryos of adequate quality obtained through multiple cycles is performed to overcome the limited number of retrieved oocytes from POR patients. However, no scientific evidence is published yet. A study was recently published on the efficacy of retrieval of oocyte through multiple hyper-stimulation in PGT [21]; however, there is no study that uses embryos to our knowledge.

Our current study focuses on the effect of the number of transferred embryos on the pregnancy rates in POR patients. The number of oocytes obtained from each retrieval was significantly higher in the fresh ET group (2.1 ± 0.8 vs. 1.6 ± 0.9, *p* = 0.00). The number of transferred embryos from ACC-ET group and fresh ET group were 3.1 ± 0.9 and 1.5 ± 0.7, respectively. However, the rate of good quality transferred embryo was significantly lower in the ACC-ET group (48.2 ± 30.5 vs. 65.4 ± 41.0, *p* = 0.00). Nevertheless, PR and CPR were significantly higher in the ACC-ET group compared to those in the fresh ET group (29.6 (21/71) % vs. 18.8 (97/517) %, *p* = 0.040; 23.5 (16/68) % vs. 14.0 (71/508) %, *p* = 0.047). Based on these results, it can be assumed that increasing the number of transferred embryos through ACC-ET would increase the pregnancy rate in POR patients.

In conclusion, when one or more oocytes cannot be retrieved from one cycle despite hyper-stimulation, retrieving multiple oocytes over the multiple cycles can also be considered. Clinical pregnancy rates (23.5% and 14% for the ACC-ET group versus the fresh ET group) in the present study were much higher than the rates of 0.9% and 8.7% previously reported for modified natural cycle IVF in POR patients according to the Bologna criteria [6,22]. Many of researchers often empirically perform ACC-ET without scientific evidence. Our analyses demonstrate that ACC-ET improves the clinical pregnancy rates and it can also be used as an alternative strategy to improve the pregnancy rates for the patients with POR. According to La Marca et al. [23], the pregnancy rate in POR patients does not exceed 8%. In the present study, LBR was greater in ACC-ET group compared to that in the fresh ET group, though the result was not statistically significant. However, LBR was higher than the previous reports among POR cases [7].

This study was limited by its retrospective nature and could not rule out embryonic factors of abortion because it did not include a chromosome study on the aborted embryos. Therefore, a significant increase in LBR might be observed in the future if evaluation of embryonic factors is possible. An ideal method to analyze the effectiveness of ACC-ET is to compare the outcome of ACC-ET with that obtained from the group in which patients underwent embryo transfer immediately after a single oocyte or embryo collection, or analyze the outcome of one embryo transfer after collecting one oocyte three times and that of ACC-ET. However, since this has been designed as a retrospective study, it was not possible to use these ideal methods for further analyses. Infertility patients usually have faith in better outcomes with a transfer of multiple embryos even at the risk of multiple pregnancy. Therefore, empirically many physicians are trying to collect more than two embryos for a single transfer. Studies on the effectiveness of ACC-ET were rare since there was only one paper recently published displaying a positive effect on the ACC-ET in POR comparing to normal responder [24]. Therefore, this study was simply focused on whether an increased number of transferred embryos resulted in improving the pregnancy rates, which we have shown in Table 3. In addition, since this study focuses on increasing the number of ET, the ACC-ET group comprises of frozen thawed ET as well as frozen thawed + fresh ET, and the two are analyzed as a group. Cost analysis was not carried out despite the fact that embryos were collected over multiple cycles. Further research into the cost factor would provide further insights into its clinical implications. POR is known to occur with aging and also the reduction of implantation rates is accompanied with increased patients’ age. In our current study, we only included patients who are under 43 and our analyses revealed that increasing the number of embryos transferred through ACC-ET would improve the pregnancy rates in POR patients who are under age 43. However, no analyses for the patients who are over 43 were included in this study. In conclusion, collecting embryos in POR patients through multiple COS or natural or modified natural methods and transferring an adequate number of embryos using frozen thawed ET would help improve the pregnancy rate.

## Figures and Tables

**Table 1 jcm-11-04940-t001:** Basal characteristics of fresh ET and ACC-ET group.

	Fresh ET (n = 517)	ACC-ET (n = 71)	*p*-Value	ACC-ET			
				Frozen		Frozen + Fresh	
				Frozen (n = 47)	*p*-Value (Compared to Fresh ET)	Frozen + Fresh (n = 24)	*p*-Value (Compared to Fresh ET)
Age (years)	38.2 ± 3.7	38.1 ± 3.5	0.88	37.7 ± 3.2	0.42	38.9 ± 4.1	0.38
BMI (kg/m^2^)	21.8 ± 3.3	21.6 ± 2.2	0.49	21.5 ± 2.2	0.54	21.8 ± 2.0	0.99
AMH (ng/mL) (±SD)	0.6 ± 0.6	0.5 ± 0.4	0.38	0.5 ± 0.4	0.52	0.5 ± 0.3	0.52
Duration of infertility (years)	5.2 ± 3.8	5.6 ± 4.2	0.42	4.9 ± 4.0	0.61	6.9 ± 4.4	0.03
FSH (mIU/mL) (±SD)	10.8 ± 9.0	11.9 ± 8.0	0.35	10.6 ± 7.5	0.91	14.4 ± 8.6	0.06
Previous IVF-ET cycles (±SD)	2.8 ± 2.4	2.6 ± 3.1	0.72	2.2 ± 2.9	0.13	3.5 ± 3.3	0.31

BMI, body mass index; AMH, Anti Müllerian Hormone; FSH, follicle-stimulating hormone; IVF-ET, in vitro fertilization-embryo transfer; ACC-ET, accumulated embryo transfer; SD, standard deviation.

**Table 2 jcm-11-04940-t002:** Comparison of oocytes and embryologic outcomes between Fresh ET and ACC-ET group.

	Fresh ET (n = 517)	ACC-ET (n = 71)	*p* Value	ACC-ET			
				Frozen		Fresh + Frozen	
				N = 47	^#^*p* Value (Compared to Fresh ET)	N = 24	^#^*p* Value (Compared to Fresh ET)
No retrieved oocyte/ET	2.1 ± 0.8	4.9 ± 2.6	0.00	5.5 ± 2.8	0.00	3.5 ± 1.1	0.00
No retrieved oocyte/retrieval	2.1 ± 0.8	1.6 ± 0.9	0.00	1.8 ± 1.0	0.025	1.25 ± 0.6	0.00
				1.8 ± 1.0		1.25 ± 0.6	0.009 ^##^
Average number of OPU attempts		3.3 (239/71)		3.4 (160/47)		3.29 (79/24)	
Maturation rate	61.1 ± 36.5	57.6 ± 28.9	0.35	63.4 ± 27.9	0.59	46.1 ± 27.8	0.017
Fertilization rate (%) (±SD)	87.6 ± 28.3	78.4 ± 22.6	0.009	73.4 ± 24.2	0.00	88.2 ± 15.3	0.90
No. of transferred embryo	1.5 ± 0.7	3.1 ± 0.9	0.00	3.3 ± 1.0	0.00	2.8 ± 0.7	0.00
Good quality embryo (%)	65.4 ± 41.0	48.2 ± 30.5	0.00	43.8 ± 30.0	0.00	56.5 ± 29.9	0.17
Cancellation rate		18.8 (45/239)		16.8 (27/160)		22.7 (18/79)	
Survival rate of Thawed embryo	-	-	-	97.1 (169/174)	-	100 (41/41)	-
Pregnancy rate	18.8 (97/517)	29.6 (21/71)	0.040	29.8 (14.47)	0.119	29.2 (7/24)	0.291
Clinical pregnancy rate	14.0 (71/508)	23.5 (16/68)	0.047	22.7 (10/44)	0.189	25.0 (6/24)	0.240
Live birth rate	9.8 (50/508)	14.9 (10/67)	0.203	16.3 (7/43)	0.277	12.5 (3/24)	0.708

ACC-ET, accumulated embryo transfer; OPU: ovum pick up. ^#^ compared to Fresh ET group; ^##^ compare between frozen and fresh + frozen ACC-ET.

**Table 3 jcm-11-04940-t003:** Comparison of IVF outcomes depending on the numbers of ET in fresh cycle.

	Fresh			*p*-Value
No. of ET	1	2	3	
Pregnancy rate	13.5 (35/259)	22.8 (43/189)	33.9 (19/56)	0.00
Clinical pregnancy rate	9.8 (25/255)	16.7 (31/186)	27.8 (15/54)	0.001
Live birth rate	6.7 (17/255)	11.3 (21/186)	22.2 (12/54)	0.003

## Data Availability

The datasets used and/or analyzed during the current study are available from the corresponding authors on reasonable request.

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
