# Peer review of "Accumulated Vitrified Embryos Could Be a Method for Increasing Pregnancy Rates in Patients with Poor Ovarian Response"

_jcm, 2022, doi:10.3390/jcm11174940_

Round 1

Reviewer 1 Report

  Regarding the structure and accuracy of the phrases, the manuscript has indeed well structured information, with supported evidence and well structured phares. Being an, observational retrospective analysis, the study followed a group of 588 patients,  which may be quite sufficient for such a study. The aim of the analysis was to evaluate the efficacy of accumulation of embryo stimulations compared to fresh embryo transfer in patients with poor ovary response under 43 years old, so again, this kind of study is of high interest. Maybe it should have been better to include patients above 43 years too.

  The manuscript is original and well defined and so, the results provide an advance in current knowledge. The results are being interpreted appropiately and are significant, as well as are the conclusions, which are, of course, supported by the results. So the article is written in an appropiate way. 

  The study is correctly designed and the analysis is being performed at high standards, so the data are robust enough to draw the conclusion. 

  Surely the paper will attract a wide readership. 

  The English language is appropiate and well understandable. 

  I only have a few things to add in the lines below:

Line 14: we aimed to assess, not „to assess”

Line 15: in patients, not „of patients”

Line 26: there were no, not „there was no”

Line 27: from our knowledge, not „to our knowledge”

Line 27: there is no clinical, not „there has been no clinical”

Line 39: without „,” after „gonadotropin”

Line 46: might obtain, not „obtain”

Line 48: who went through, not „who have gone through”

Line 52: alternative methods, not „alternatives methods”

Line 96: were obtained, not „are obtained”

Line 103: „,” after „cycle”

Line 125: patients’ characteristics, not „patient characteristics”

Line 131: oocytes, not „oocyte”

Line 135: good quality transferred embryos, not „good quality embryos transferred”

Line 137: greater number of transferred embryos, not „greater number of embryo transferred”

Line 149: patients’ characteristics, not „patient characteristics”

Line 163: retrieved oocytes, not „retrieved oocyte”

Line 179: „,” before „would”

Line 187: no study yet, not „no study of yet”

Line 187: results, not „result”

Line 195: was published, not „has been published”

Line 195: a study was recently published, not „a study has recently been published”

Line 216: no spaces between [23] and „,”

Line 219: the previously, not „that previously”

Line 225: are analysed, not „are analyses”

Line 232: were not included, not „have not been included”

Reviewer 2 Report

Paper JCM 1842239 by Shin et al.

·      Accumulated vitrified embryos could be a method for increasing pregnancy rates in patients with poor ovarian response

1.    Overall opinion

This paper aims to check if accumulation of oocytes / embryos can be an efficient option to increase the chances of pregnancy in poor responders. However, this paper suffers from a very important methodological problem. Comparing a single fresh cycle (the FRESH-ET group) with a group with 3 fresh OPU in average evidently favours the second group since more oocytes, embryos and transferred embryos could be obtained. To justify the accumulated vitrified embryos strategy, it would be necessary to compare 2 groups with the same number of aspirations, one in which embryos are accumulated before transfer, and one in which embryos (if available) were transferred at each aspiration. Moreover, the 3 groups are very different (table 2) in term of number of retrieved oocytes per aspiration, of total number of transferred embryos, of good quality embryos. These differences also exist between the 2 subgroups of ACC-ET.

Thus, the conclusions are totally flawed by the groups characteristics differences and by the groups definition. It is a pity since the question is interesting: is it better to accumulate embryos during 3 aspiration cycles and to make one transfer after vitrification, or to make the same number of aspirations with transfer after each. The paper does not answer that question. The only answer from this paper is that performing 3 aspirations (with subsequent vitrification) gives higher PR than a single one !!.

Moreover, if authors submit again this paper with appropriate changes, it will be useful to compare cumulative rates (including potential FET after fresh transfer and 2nd FET in the ACC-ET group if any)

2.    Recommendations

This paper needs major revision

3.    Detailed comments

a.    Introduction

Introduction is ok but is based on relatively old literature (1996-2014; only 1 from 2015 and 1 from 2018).

b.    Material and methods

The definition of POR is ok, as the stimulations and FET protocols and the statistical methods

c.     Results:

Results are correctly reported, but as underlined in the general opinion, it is not that section which is controversial

d.    Discussion:

This section does not discuss the major cause of bias secondary to the protoco
